# GnRH Vaccine Could Suppress Serum Testosterone in Stallion Mules

**DOI:** 10.3390/ani14121800

**Published:** 2024-06-17

**Authors:** Siriporn Khumsap, Sahatchai Tangtrongsup, Patcharapa Towiboon, Chaleamchat Somgird

**Affiliations:** 1Equine Clinic, Faculty of Veterinary Medicine, Chiang Mai University, Chiang Mai 50100, Thailand; 2Small Animal Clinic, Faculty of Veterinary Medicine, Chiang Mai University, Chiang Mai 50100, Thailand; sahatchai.t@cmu.ac.th; 3Research Center of Producing and Development of Products and Innovations for Animal Health and Production, Faculty of Veterinary Medicine, Chiang Mai University, Chiang Mai 50100, Thailand; 4Center of Elephant and Wildlife Health, Faculty of Veterinary Medicine, Chiang Mai University, Chiang Mai 50100, Thailand; towiboon@gmail.com (P.T.); chaleamchat.s@cmu.ac.th (C.S.); 5Elephant and Wildlife Clinic, Faculty of Veterinary Medicine, Chiang Mai University, Chiang Mai 50100, Thailand

**Keywords:** stallion mule, immunocastration, GnRH vaccine, Improvac, anti-GnRH antibody, testosterone

## Abstract

**Simple Summary:**

Stallion mules are infertile but still able to produce testosterone, which influences undesirable behavior. Surgical castration is required to eliminate testosterone synthesis, which results in pain and the risk of postoperative complications. To the best of our knowledge, there are no existing studies regarding immunocastration as an alternative method for surgical castration in mules. The objective of this study was to evaluate whether a gonadotropin-releasing hormone (GnRH) vaccine could be used as a method of immunocastration in stallion mules via the assessment of anti-GnRH antibody concentration, serum testosterone concentration, clinical adverse effects, and changes in the selected behaviors. Intact and unilateral cryptorchid mules received the GnRH vaccine at weeks 0, 4, and 8. The serum testosterone concentrations in GnRH-vaccinated mules were lower than before vaccination from weeks 6 to 14. Subcutaneous edema adjacent to the injection site was observed in intact mules after the second or third vaccination. The stallion mules responded to the GnRH vaccine, resulting in a temporary decrease in serum testosterone. Based on the results of this study, the GnRH vaccine may be administered as a temporary immunocastration method for stallion mules.

**Abstract:**

Stallion mules have been used as working equids in several countries. Aggressiveness under the influence of testosterone results in the necessity for surgical castration before work training. The gonadotropin-releasing hormone (GnRH) vaccine may be an alternative method for immunocastration in mules. The objective of this study was to evaluate the effect of the GnRH vaccine on anti-GnRH antibody concentration, serum testosterone concentration, clinical adverse effects, and behavioral changes in response to receiving selected physical manipulations from humans. Twenty-five mules were separated into three groups: Control-intact, Control-castrated, and Treatment. The Treatment group was further divided according to condition (intact or unilateral cryptorchid) and age. The Treatment group received 195 µg of the GnRH vaccine intramuscularly at weeks 0, 4, and 8. The anti-GnRH antibody concentrations increased at weeks 6 and 10, and then they gradually decreased to baseline at week 24. The Treatment-intact-young group had the highest concentration of anti-GnRH antibody. The serum testosterone concentrations in the Treatment group were lower than before vaccination from weeks 6 to 14. Subcutaneous edema adjacent to the injection site was detected in the Treatment-intact group after booster vaccination. In conclusion, the mules responded to the GnRH vaccine, which could temporarily suppress testosterone for up to 14 weeks.

## 1. Introduction

The mule, a hybrid between a donkey (jack) and a female horse (mare), has been used for several types of work, including military tasks, heavy industry work, and recreational and sports activities. Mules are intelligent, not naturally aggressive, and are preferred for military work in the mountains over horses and donkeys [1,2]. Despite being infertile, the stallion mules exhibit normal testosterone synthesis, which may influence the stallion’s aggressiveness or other undesirable behavior towards humans [3,4]. Therefore, stallion mules are usually castrated before work training. During routine castrations, both testes are surgically removed from the scrotal sacs, and the wounds are generally left open to drain and heal by secondary intention. The incidence of perioperative and postoperative complications ranges from 10.0% to 60.0% according to the surgical procedure or postoperative management [5,6,7,8,9,10,11]. Retained abdominal testes can produce testosterone in cryptorchid stallions [12], which may contribute to undesirable behavior. Laparotomy or laparoscopic surgery must be performed to remove the abdominal testes from cryptorchid animals. The procedure requires specialized facilities and expensive equipment, which are usually available in referral hospitals. In addition, there is a possible risk of postoperative complications. Novel methods to reduce testosterone concentration in equids, apart from surgery, may provide an alternative choice for the owners. Based on the results of previous studies, vaccination against gonadotropin-releasing hormone (GnRH) appeared to be the most suitable immunocastration method for clinical use in horses [13,14]. Additionally, the GnRH vaccine is recommended as a method to achieve a balance between animal welfare and the reduction in undesirable behaviors due to sex hormones in working animals [15]. In Thailand, one of its military units is responsible for the production of mules for military-specific tasks around the mountain border. Due to some limitations on surgical management and the regular schedule to transfer mules from the producing unit to the training unit, some mules were left uncastrated. Consequently, a group of intact and cryptorchid mules remained within the training unit, which caused management difficulties during pasture turn-out and training. The unit solution was to confine that group of mules to either a stall or a limited area, which might affect these animals’ mental or overall well-being. In order to assist in such a situation, our primary interest was to evaluate whether the GnRH vaccine could be used as an immunocastration method in stallion mules. To the best of our knowledge, immunocastration using the GnRH vaccine has not been previously studied in mules. Because humane surgical castration for mules may be difficult in some countries, the findings of this study may apply to such situations. The only form of the GnRH vaccine available in Thailand is a vaccine licensed for male pigs. Nonetheless, this product has been used in several studies in horses [16,17,18,19] and should be applicable to the study of intact and cryptorchid mules. The objective of this study was to evaluate the effect of the GnRH vaccine in mules on anti-GnRH antibody concentration, serum testosterone concentration, clinical adverse effects, and behavioral changes in response to receiving selected physical manipulations from humans.

## 2. Materials and Methods

This study was approved by the Animal Care and Use Committee, Faculty of Veterinary Medicine, Chiang Mai University (R17/2561). The period of this study was from the beginning of May to the end of October. The protocols presented in Section 2.2 and Section 2.3 were performed between 8.30 and 11.30 AM for all participating mules throughout the period of this study.

### 2.1. Animals

The study involved a total of 25 mules from a military unit in northern Thailand, including 13 intact stallion mules, 6 unilateral cryptorchid mules, and 6 castrated mules. The intact stallion mules were confirmed via palpation and detection of both testes in the scrotum. The inclusion of unilateral cryptorchid mules was based on one palpable testis in the scrotum and an impalpable testis or mass in the inguinal area on the opposite side. The exact locations of the retained testes could not be determined. Two intact stallion mules were randomly selected for the Control-intact group. The remaining intact and cryptorchid mules were assigned to the Treatment group, which was further divided into subgroups according to condition (intact or unilateral cryptorchid) and age as follows: (a) Treatment-intact-young group (age 5–9 years, *n* = 4); (b) Treatment-intact-old group (age 10–15 years, *n* = 7); and (c) Treatment-cryptorchid group (age 5–15 years, *n* = 6). Six mules identified as castrated mules in the official list of the military unit were randomly selected from the same herd as the Control-castrated group: three mules aged 5–9 years and three mules aged 10–15 years. The precise surgical history of each mule was unknown. Palpation was used to confirm the absence of testes in both inguinal areas for this particular group.

### 2.2. GnRH Vaccination and Clinical Examination for Adverse Effects

The week of the first vaccination was indicated as week 0. The Treatment group received 195 µg (1.3 mL) of the GnRH vaccine (Improvac, Zoetis, Berlin, Germany, 150 µg/mL) intramuscularly at the neck at weeks 0, 4, and 8. The first and third injections were administered on the left side, whereas the second injection was administered on the right side. The Control-intact and Control-castrated groups received 1.3 mL of 0.9% NaCl solution using the same method described for the Treatment group. A clinical examination for adverse effects was performed prior to vaccination and daily for three days after vaccination, including rectal temperature (°C), injection site swelling (score 0 to 2), and pain at the injection site with hand pressing three times (yes/no), as described in a previous study [19]. In brief, a score of 0 indicated no swelling. The higher score indicated an increase in the severity of adverse effects. A rectal temperature exceeding 39 °C indicated a fever. If any adverse effects occurred within one to three days after vaccination, the animal would receive the relevant treatment. For animals with fever, phenylbutazone at a dosage of 4.4 mg/kg intravenously would be applied. For animals with injection site swelling, a 0.5% piroxicam topical anti-inflammatory gel would be applied to the swelling area. The treatment would be administered on a daily basis until the adverse effects subsided.

### 2.3. Samples and Data Collection

All the mules in this study underwent training to be led by hand using a halter and a lead rope. Each mule had the experience of being confined in the restraint stock at least twice a year for the military’s annual health check and blood collection program. At least one day before the data collection, the mules were housed in individual stalls in two similar stables. On the day of data collection, each mule was fitted with a halter and a lead rope and allowed to rest and graze for approximately 5–10 min in the area adjacent to the restraint stock, which was adjacent to each stable. The mule was then led by the military staff into the restraint stock, where samples and data were collected.

Blood samples and video recordings were collected from each mule every two weeks, starting from two weeks (week 2) before the first vaccination until week 24. A 10 mL blood sample was collected in a plain tube and kept in an ice box during transportation. Serum was separated after centrifugation and kept at −20 °C until analysis. Video recording was continuously conducted, starting from the introduction of each mule into the restraint stock until the end of all data collection procedures. One person, unfamiliar with the animals and uncertain whether mules were included in the Treatment or Control groups, assessed the video recordings throughout the study. The mule’s behaviors within the restraint stock were assessed in three aspects: (a) response when each mule stood still for one minute without human interaction; (b) response to palpation at the scrotal area; and (c) response when blood collection was performed. The behavioral score was graded from 0 to 3 as follows: 0 (at ease: calm, carefree), 1 (annoyed: inability to stand still, vigilant), 2 (alarmed: inability to stand still, ears flat-back against the head, highly vigilant, or pawing), and 3 (aggressive: attempts to escape from the restraint stock, attacking, or additional nose or ear twist required to fulfill data collection tasks).

At weeks 0, 12, and 24, total scrotal width (TSW) was measured three times from each intact mule using a stallion scrotal caliper. In the same week, body weight (BW) was measured three times from each mule using an equine weight measuring tape.

### 2.4. Serum Anti-GnRH Antibody Analysis

The anti-GnRH antibody concentration was analyzed using direct non-competitive ELISA. The protocol for plate preparation was similar to a previous study [19]. After plate blocking with protein buffer (40 mM Na_2_HPO_4_, 0.68 M NaCl, 13.4 mM KCl, 7.35 mM KH_2_PO_4_, 5% *w*/*v* Na Casien, 0.25% *v*/*v* Tween 20, 0.05% *w*/*v* thiomersal; pH 7.0) and one-hour incubation at room temperature (RT), the plates were washed five times with washing solution (8.1 mM Na_2_HPO_4_, 0.14 M NaCl, 2.68 mM KCl, 1.47 mM KH_2_PO_4_, 0.05% *v*/*v* Tween; pH 7.0); loaded with 100 µL each of the control (from an animal vaccinated with Improvac, Zoetis, Germany), diluted serum samples (1:50–1:1500), and high- and low-concentration controls; and then incubated for two hours at RT. Next, the plates were washed five times, and then 100 µL of conjugate (protein G Peroxidase, P8170, Sigma Chemical Co., St. Louis, MO, USA) was added at 1:20,000 dilution in buffer (40 mM Na_2_HPO_4_, 0.68 M NaCl, 13.4 mM KCl, 7.35 mM KH_2_PO_4_, 0.25% *v*/*v* Tween 20, 0.05% *w*/*v* thiomersal; pH 7.0) and incubated for one hour at RT. After washing, 100 µL of 3,3′,5,5′-tetramethylbenzidine dihydrochloride (TMB, Sigma, USA, T3405) dissolved in phosphate–citrate buffer with sodium perborate was added (Sigma, USA, P4922), followed by incubation for 20 min at RT. The reaction was stopped with 50 µL of stop solution (2 M H_2_SO_4_), and absorbance was measured at 450 nm by using a microplate reader (Sunrise, Tecan, Männedorf, Switzerland). The inter-assay coefficient of variation (CV) for the high- and low-concentration controls was less than 10%. The intra-assay CVs were less than 10%. Data were compared with a standard curve to identify the anti-GnRH antibody concentration (unit).

### 2.5. Serum Testosterone Analysis

The serum testosterone concentration was analyzed using a double-antibody enzyme immunoassay as previously described [20] with minor modifications. The concentration of two substrates was modified: steroid horseradish peroxidase conjugate (HRP) at 1:20,000 and testosterone antibody at 1:110,000. Absorbance was measured at 450 nm by using a microplate reader (Sunrise, Tecan, Switzerland). The sensitivity of the assay was 0.0714 ng/mL. The inter-assay CV for the high- and low-concentration controls was less than 10%. The intra-assay CVs were less than 10%. The serum testosterone concentration was reported as ng/mL.

### 2.6. Statistical Analysis

One castrated mule was indicated as an outlier. Due to its highly distinct basal testosterone concentration, the data of this mule were excluded from statistical analysis. The statistical analysis was performed based on 24 mules separated into the following three groups: (a) Treatment group (*n* = 17), which was further divided into three subgroups (Treatment-intact-young group, *n* = 4; Treatment-intact-old group, *n* = 7; and Treatment-cryptorchid group, *n* = 6); (b) Control-intact group (10–15 years old, *n* = 2); and (c) Control-castrated group (*n* = 5). In week 16, eight mules were included in the annual military training event. Consequently, the data from those mules were missing for that particular week, including two mules in the Treatment-intact-young group, two mules in the Treatment-intact-old group, and four mules in the Treatment-cryptorchid group.

All quantitative dependent variables were tested for normality using the Shapiro–Wilk test. The testosterone reduction ratio of the specific week (*W_i_*) was calculated for the Treatment group by using the serum testosterone concentration at week 0 (*W*_0_) as a baseline and was calculated as follows:$$Testosterone\;reduction\;ratio\;of\;W_{i}=\frac{\left(Testosterone\;W_{0}-Testoserone\;W_{i}\right)}{Testosterone\;W_{0}\;}$$

Multiple comparisons between groups of anti-GnRH antibody concentrations, serum testosterone concentrations, TSW, and BW were analyzed using Kruskal–Wallis or one-way ANOVA tests, whichever was appropriate. Due to the limitations of the video recordings at week 0, the behavioral scores at week 2 were selected as a baseline for comparison with scores from other weeks. Friedman and Dunn’s tests with Bonferroni adjustment were used to determine differences in the mean of anti-GnRH antibody concentrations, serum testosterone concentrations, testosterone reduction ratio, TSW, BW, and behavioral scores over time. The serum testosterone profiles before vaccination were analyzed via descriptive analysis using pooled serum testosterone concentrations at weeks −2 and 0 from 24 mules. The Mann–Whitney U test was used to analyze the differences in serum testosterone concentrations between young (*n* = 4) and old (*n* = 9) intact mules, between intact (*n* = 13) and unilateral cryptorchid (*n* = 6) mules, and between young intact (*n* = 4) and young cryptorchid (*n* = 4) mules. Clinical adverse effects were analyzed via descriptive analysis. All statistical analyses were performed using the Stata statistical software release 16.1 (StataCorp., College Station, TX, USA).

## 3. Results

### 3.1. Anti-GnRH Antibody Concentration

In the Treatment group (*n* = 17), the anti-GnRH antibody concentrations started to rise two weeks after the first vaccination (Figure 1). Based on multiple comparisons, there were significant differences (*p* < 0.05) in the anti-GnRH antibody concentrations between the Treatment group and both Control groups from weeks 4 to 24 (Figure 1).

Multiple comparisons of anti-GnRH antibody concentrations between the Treatment subgroups indicated a significant difference (*p* < 0.05) between the Treatment-intact-young and Treatment-intact-old groups from weeks 2 to 24, except for week 16 (Figure 2). The anti-GnRH antibody concentrations of the Treatment-cryptorchid group were not significantly different (*p* > 0.05) from those of the Treatment-intact-young and Treatment-intact-old groups. The highest anti-GnRH antibody concentration was determined in the Treatment-intact-young group at week 10 (66,864 ± 20,775 units; mean ± SEM), as shown in Figure 2.

### 3.2. Serum Testosterone Concentration

The serum testosterone profiles of all intact stallion mules (*n* = 13) prior to vaccination were 6.06 ± 1.83 ng/mL (range 0.07–36.61 ng/mL), shown as means ± SEM. When divided by age, the serum testosterone profiles in young intact stallion mules (*n* = 4) were 3.08 ± 1.43 ng/mL (range 0.12–11.88 ng/mL), and in old intact stallion mules (*n* = 9) they were 7.39 ± 2.54 ng/mL (range 0.07–36.61 ng/mL). The serum testosterone profiles of all cryptorchid mules (*n* = 6) were 4.19 ± 2.66 ng/mL (range 0.07–26.76 ng/mL). When divided by age, the serum testosterone profiles in young cryptorchid mules (*n* = 4) were 0.35 ± 0.09 ng/mL (range 0.07–0.72 ng/mL), and in old cryptorchid mules (*n* = 2) they were 11.88 ± 6.93 ng/mL (range 0.07–26.76 ng/mL). The serum testosterone profiles of the Control-castrated group (*n* = 5) were 0.37 ± 0.18 ng/mL (range 0.07–1.78 ng/mL). The serum testosterone concentrations between young and old intact stallion mules were not statistically different (*p* > 0.05). There were statistically significant differences (*p* < 0.05) in serum testosterone concentrations between intact and cryptorchid mules, as well as between young intact and young cryptorchid mules.

Multiple comparisons of serum testosterone concentrations between both Control groups and the three Treatment subgroups are shown in Figure 3. Due to the high variation in serum testosterone concentrations in some individuals, there was no statistical difference (*p* > 0.05) between those five groups at any time point. At week 0, there was a trend toward a significant difference (*p* = 0.06) in serum testosterone concentrations between the Treatment-intact-young and Control-castrated groups. During weeks 2 to 24, the concentrations of these two groups were not significantly different (*p* > 0.05). The statistical analysis of the testosterone reduction ratio of the Treatment group (*n* = 17) indicated significant differences (*p* < 0.05) in serum testosterone concentrations at weeks 6, 10, 12, and 14 when compared with the concentration at week 0.

### 3.3. Behavioral Response, Changes in Total Scrotal Width, and Body Weight

The comparisons of the behavioral scores at weeks 6, 12, 18, 22, and 24 with the score at week 2 are presented in Table 1. The behavioral scores of both Control groups were not significantly different (*p* > 0.05) throughout the study period, while the score of response to blood collection activity of the Treatment group significantly decreased (*p* < 0.05) from weeks 6 to 24.

At week 0, the TSW of all intact mules in the Treatment group (*n* = 11) was 6.5 ± 0.3 cm, shown as means ± SEM. For the Treatment-intact-young (*n* = 4) and Treatment-intact-old (*n* = 7) groups, they were 7.0 ± 0.4 cm and 6.0 ± 0.4 cm, respectively. The TSW of the Treatment-intact-young group indicated a trend toward significant decreases (*p* = 0.09) at week 12 (mean difference −1.0 cm) and week 24 (mean difference −1.5 cm), whereas the TSW of the Treatment-intact-old group was not significantly different (*p* = 0.3) from week 0.

The means ± SEM of BW at week 0 from the participating mules were as follows: Control-intact group, 218 ± 14 kg; Control-castrated group, 273 ± 14 kg; Treatment-intact-young group, 246 ± 7 kg; Treatment-intact-old group, 232 ± 10 kg; and Treatment-cryptorchid group, 261 ± 11 kg. The comparisons of BW at week 0 with weeks 12 and 24 showed no statistical difference (*p* > 0.05) in all groups.

### 3.4. Clinical Adverse Effects

The occurrence of clinical adverse effects in the Treatment group (*n* = 17) is shown in Table 2. After the first and second vaccinations, three and six mules developed a fever on the subsequent day. After the third vaccination, one mule (no. 6) developed a fever on the subsequent day, while three mules (nos. 7, 11, and 14) developed a fever two days after vaccination. The affected mules received phenylbutazone intravenously as previously described. The body temperatures were below 39 °C on the subsequent day after the treatment. After the first vaccination, only one mule (no. 16) developed a small injection site swelling without pain on the subsequent day. The swelling was pronounced after the booster vaccination. On the third day following the second vaccination, large subcutaneous edema—ranging in size from 3 to 6 cm—developed around or beneath the injection site in three mules (nos. 9, 14, and 16). Similar side effects occurred in seven mules (nos. 1, 6, 7, 11, 13, 16, and 17) on the subsequent day after the third vaccination. No heat or pain response was detected in these mules based on hand pressing at the injection site or edematous area. The subcutaneous edema subsided after the daily application of topical anti-inflammatory gel to the affected area for a few days. After a total of three vaccinations, different clinical adverse effects were observed in 10/11 intact stallion mules. There were minor clinical adverse effects in 4/6 cryptorchid mules. Large subcutaneous edema was observed only in the intact stallion mules, comprising 100% of the young group (5–9 years, four affected mules) and 71.43% of the old group (10–15 years, five affected mules).

## 4. Discussion

The GnRH vaccine has been evaluated for immunocastration in stallion horses [14,21,22,23,24] and stallion donkeys [25]. To the best of our knowledge, our study is the first to report information on the GnRH vaccination in stallion mules. Mules responded to the GnRH vaccine in a similar pattern to that of stallion horses, in which the anti-GnRH antibody titers started to rise after the second vaccination and reached peak values around 2–4 weeks after the third vaccination [21]. The anti-GnRH antibody concentrations in the Treatment-intact-young group were higher when compared to those in the Treatment-intact-old group. Similar findings were observed in the previous study, in which the anti-GnRH antibody titers in the young mares (four years or less) were higher than in the older mares (11 years or more) [17]. This finding could be related to immunosenescence, which is the decrease in immune system ability that occurs in older animals [14,26]. In our study, evidence of hypothalamic–pituitary–testicular axis impairment due to the GnRH vaccine was detected at least in the Treatment-intact-young group. This group had the highest anti-GnRH antibody concentration, resulting in the declination of serum testosterone concentrations within two weeks after the first vaccination. The duration was comparable to the 15 days of testosterone depletion after surgical castration in young horses [27]. Consequently, this group of mules had a trend toward reductions in total scrotal width at weeks 12 and 24. A similar effect on testes was observed in stallion horses treated with the GnRH vaccine [22,23]. The testicular volume in stallions decreased by 70% at 90 days after vaccination due to the reduction in seminiferous tubular tissue relative to interstitial tissue [22]. A reduction in scrotum size of around 37% was observed in stallions three months after the first GnRH vaccination [23].

In our study, the serum testosterone concentrations of stallion mules were highly varied, both within and between groups. Several factors affect serum testosterone concentration in individual equids, such as age [28], season and photoperiod in the temperate region [28,29], infertility [30], and housing system [31]. Three mules in our study—two in the Treatment-intact-old group and one in the Treatment-cryptorchid group—had distinctly high basal serum testosterone concentrations. Age influences testosterone concentrations in stallion horses from birth to early puberty [28] but not from middle age (8–14 years old) to old age (more than 15 years old) [29]. The age of mules in our study ranged from 5 to 15 years old; therefore, the effect of age should be minimal. The mean testosterone concentrations of stallion horses in a tropical region did not differ among seasons due to the minimal changes in day length during the year [32]. Thailand is in a similar region (15° latitude region) as in a previous study (20° latitude region) [32]. Therefore, changes in serum testosterone concentrations in our study should have minimal effect from seasonal change. In the previous studies [12,30], infertile and non-breeding stallion horses had lower serum testosterone concentrations than those in the breeding stallion horses. Mules are born infertile; nonetheless, there is evidence of fertile female mules [33]. The stallion mules may resemble the sexual status identified as infertile, non-breeding, or breeding as in the stallion horses, which may affect the concentrations of serum testosterone in each sexual status. The housing system influences the basal testosterone concentrations in stallion donkeys [31]. The testosterone concentrations were higher when stallion donkeys were housed in individual stalls as compared to being housed together in a paddock. In our study, the military’s mule herd was usually housed together in a small paddock at night and released to the pasture in the daytime. Mules causing trouble in the herd were housed in individual stalls most of the day. The information regarding the usual housing of all the mules who participated in our study was not known. Therefore, the variation in serum testosterone concentrations of individual mules in our study may be influenced by at least two factors: sexual status and the housing system. Multiple comparisons between groups with high variation may have the least meaning and may be difficult to interpret. In our study, the analysis of testosterone reduction ratio in the Treatment group provided a meaningful finding to be discussed.

The serum testosterone concentrations in the Treatment group significantly decreased within two weeks after the second vaccination. Similar results were observed in stallion horses where the serum testosterone concentrations decreased within two months after receiving the GnRH vaccine, two doses 4–6 weeks apart [24]. However, the period of testosterone suppression in that study was more extended (5–6 months) than in our study. In stallion donkeys, the serum testosterone concentrations decreased at day 60 after receiving the GnRH vaccine, two doses 30 days apart [25]. The serum testosterone concentrations in the stallion donkeys remained low for 120 days, similar to the duration of testosterone suppression observed in the Treatment group of our study. It is unknown whether the serum testosterone concentrations of stallion donkeys beyond 120 days would be maintained at a low concentration or rise as mules’ testosterone concentrations did in our study. The variation in duration of testosterone suppression across species may be due to physiological differences within each type of equid, as inter-individual variation against the GnRH vaccine occurred even within the same species [21,23].

In the previous study [34], the mean serum testosterone concentrations of twelve young stallion mules, separated into four groups, were 2.12, 2.05, 2.07, and 2.05 ng/mL. The mean basal testosterone concentration in young intact mules in our study was slightly higher than that. The previous study was conducted in the temperate region, where seasonal changes and photoperiods affected serum testosterone concentration [28,29]. If that study was conducted in the non-breeding season, the testosterone concentration might be lower than in the breeding season. In addition, testosterone secretion shows a diurnal pattern [14]. There was no information regarding the month or the period during the day in which the previous study was conducted. The slight difference in serum testosterone concentrations between the previous study and ours may be due to research conducted in different periods of the year, different regions of the study (temperate vs. tropical), different times of blood collection during the day, as well as different laboratory techniques.

In stallion horses, the serum testosterone concentrations increase with age but become stable after puberty at six years old [29]. This might explain why young and old intact mules in our study had similar serum testosterone concentrations. In horses, the serum testosterone concentrations between cryptorchid and intact stallions were similar [35]. In that study, the retained testes of cryptorchid horses were mostly located in the inguinal canal. Our study revealed the differences in serum testosterone concentrations between unilateral cryptorchid and intact mules. The retained testes of cryptorchid mules in our study were assumed to be located in the abdomen. The authors of [36] stated that the retained abdominal testis may contribute to some negative effects on the function of the scrotal testis in cryptorchid stallion horses. A recent study indicates a lower testosterone concentration in cryptorchidism when compared to stallion horses [37]. In addition, the serum testosterone concentrations in cryptorchid horses increase from one to five years of age, followed by a gradual decrease [38]. Chronic exposure of the abdominal testis to high temperatures affects Leydig cell function, resulting in a decrease in testosterone synthesis and an increase in the conversion of androgens into estrogens [37]. The age of cryptorchid mules in our study ranged from 5 to 15 years old, in which the aging may be associated with low serum testosterone concentrations, as demonstrated in cryptorchid horses. Although male equids share some reproductive similarities [39], a few differences may be considered. The different findings between horses in the previous study [35] and mules in ours may also be due to the physiological differences between equids, locations of the retained testes, the number and age of animals in the study, or other factors. Further investigation may be needed to confirm the serum testosterone profiles using larger groups of cryptorchid and intact mules.

In the intact stallion horses, the owners subjectively reported that the GnRH vaccine reduced distraction and aggressive behavior [16]. The effect of the GnRH vaccine on behavioral changes in our study was statistically indicated in at least one aspect. In the Treatment group, the initial changes in response to blood collection activity corresponded with the initial decrease in serum testosterone concentrations. The behavioral scores of both Control groups were not changed in any aspect, which may indicate the consistency of the mules’ behaviors. Although the behavioral scores for blood collection subjectively decreased in all groups, it is possible that the low number of mules in both Control groups did not allow for statistical differences due to low power statistics. Therefore, the changes in behavioral scores in our study may or may not be associated with the progressive acclimatization of the animals to the study protocols. The consistency of behavioral scores over time may relate to inherent behaviors and may not be directly influenced by testosterone. Mules are intelligent and can avoid potentially dangerous situations. The military mules may have learned from previous experiences that being positioned in the restraint stock for veterinary procedures results in painful sensations. This may explain why the mules in our study were mostly graded as scores 1 to 2 when they were left alone to stand still for one minute in the restraint stock. Some mules were graded as score 3 due to the attempt to escape from the restraint stock. Although the statistical analysis indicated significant behavioral changes toward ease of blood collection activity in the Treatment group, the decrease in mean score was small and may not be recognized in a real situation.

In our study, the GnRH vaccine at a dosage of 195 µg was selected based on suppression effects on sex hormone production and adverse effects from previous studies in horses [16,21,40,41,42]. In the studies using a licensed GnRH vaccine for horses (Equity) at a dosage of 200 µg, stallion horses did not develop fever or any clinical adverse effects [16,21], but visible swelling developed in mares after the second vaccination [40]. Conversely, horses receiving a licensed GnRH vaccine for pigs (Improvac) at a dosage of 400 μg developed several adverse reactions, including swelling and pain at the injection site, stiffness of the neck, pyrexia, apathy [41], and lameness [42]. Although the dosage of the GnRH vaccine in our study was comparable to the previous studies [16,21], mules in the Treatment group still developed some adverse effects, especially after the booster dose. The more severe adverse effects were also observed in horses receiving booster doses of the GnRH vaccine (Improvac) at a dosage of 400 µg [41]. Stallion donkeys did not develop any clinical adverse effects after receiving a licensed GnRH vaccine for cattle (Bopriva) at a dosage of 400 µg twice [25]. Other compositions of chemicals and adjuvants within different brands of vaccines may contribute to the occurrence of clinical adverse effects. The development of large subcutaneous edema after booster vaccination may relate to hypersensitivity type III or Arthus reaction [43]. Immune complex formation within or around blood vessels at the injection site triggered the inflammatory process, resulting in local swelling and pain. The reaction of mules in this study might relate to the alteration in vascular permeability, resulting in fluid leakage from capillaries to the subcutaneous space around the injection site. The reason this reaction occurred only in the intact mules and not in the cryptorchid mules is unclear. It might relate to the concentration of the anti-GnRH antibody and the amount of antigen–antibody complex formation. The edematous blemish at the neck of mules in our study might not influence daily grazing and feed consumption, as the BW of the Treatment group did not differ between weeks 0, 12, and 24.

## 5. Conclusions

Both intact and cryptorchid mules responded to the GnRH vaccine, resulting in measurable anti-GnRH antibodies in serum. The young intact stallion mules had the highest antibody concentration, resulting in decreases in serum testosterone concentrations and total scrotal width compared to before vaccination. Based on the results of this study, the GnRH vaccine may be administered as a temporary immunocastration method in stallion mules. Daily body temperature measurement and observation of the injection site should be carried out for at least three days after vaccination. Furthermore, relevant medication should be provided if adverse effects occur.

## Figures and Tables

**Figure 1 animals-14-01800-f001:**
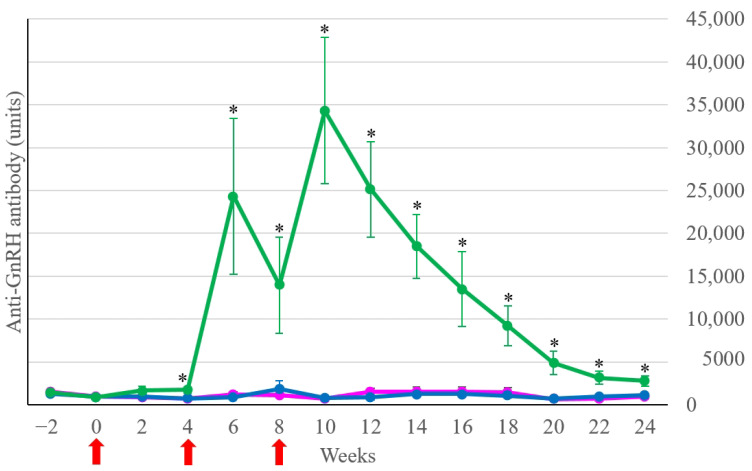
Anti-GnRH antibody concentrations (units) of the Treatment group (green line), the Control-intact group (dark blue line), and the Control-castrated group (pink line). Lines and bars indicate the mean ± SEM of anti-GnRH antibody concentrations at each time point. The red arrows denote the vaccination time. * *p* < 0.05 between the Treatment and both Control groups.

**Figure 2 animals-14-01800-f002:**
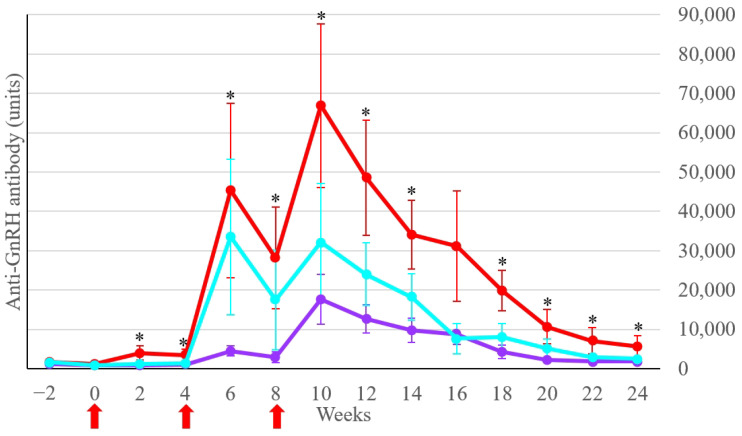
Anti-GnRH antibody concentrations (units) of the Treatment subgroups: Treatment-intact-young group (red line), Treatment-intact-old group (purple line), and Treatment-cryptorchid group (blue line). Lines and bars indicate the mean ± SEM of anti-GnRH antibody concentrations at each time point. The red arrows denote the vaccination time. * *p* < 0.05 between the Treatment-intact-young and Treatment-intact-old groups.

**Figure 3 animals-14-01800-f003:**
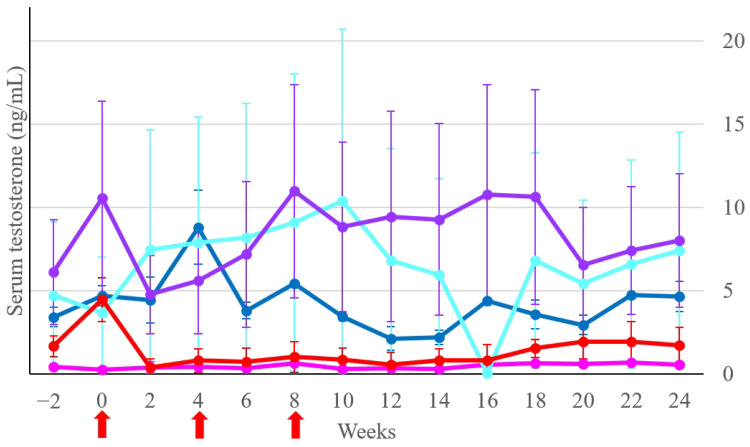
Serum testosterone concentrations (ng/mL) of the Control-intact group (dark blue line), the Control-castrated group (pink line), and the Treatment subgroups: Treatment-intact-young group (red line), Treatment-intact-old group (purple line), and Treatment-cryptorchid group (blue line). Lines and bars indicate the mean ± SEM of serum testosterone concentrations at each time point. The red arrows denote the vaccination time.

**Table 1 animals-14-01800-t001:** Comparison of the behavioral scores (0, 1, 2, or 3) between week 2 and weeks 6, 12, 18, 22, and 24. The numbers indicate the mean scores (ranges of scores) of the indicated groups. * indicates *p* < 0.05 when comparing behavioral scores in the particular week with the scores in week 2.

Group	Week 2	Week 6	Week 12	Week 18	Week 22	Week 24
Control-intact group (*n* = 2)						
Stood still for one min	1.0 (0–2)	1.5 (0–3)	1.5 (1–2)	0.5 (0–1)	0.5 (0–1)	0.0 (0–0)
Palpation at scrotal area	1.0 (0–2)	1.5 (1–2)	0.5 (0–1)	0.5 (0–1)	0.5 (0–1)	1.5 (0–3)
Blood collection	1.0 (0–2)	0.0 (0–0)	0.5 (0–1)	0.0 (0–0)	0.0 (0–0)	0.0 (0–0)
Control-castrated group (*n* = 5)						
Stood still for one min	0.5 (0–1)	2.0 (1–3)	1.0 (1–1)	1.5 (0–3)	1.0 (0–3)	0.8 (0–3)
Palpation at scrotal area	0.3 (0–1)	0.8 (0–1)	0.3 (0–1)	0.0 (0–0)	0.0 (0–0)	0.8 (0–3)
Blood collection	0.8 (0–1)	0.3 (0–1)	0.3 (0–1)	0.0 (0–0)	0.0 (0–0)	0.0 (0–0)
Treatment group (*n* = 17)						
Stood still for one min	1.1 (0–2)	0.8 (0–2)	1.0 (0–2)	1.3 (0–2)	0.6 (0–2)	0.7 (0–2)
Palpation at scrotal area	0.7 (0–2)	0.3 (0–1)	0.2 (0–1)	0.1 (0–1)	0.0 * (0–0)	0.5 (0–2)
Blood collection	0.7 (0–2)	0.1 * (0–1)	0.0 * (0–0)	0.1 * (0–2)	0.0 * (0–0)	0.3 * (0–3)

**Table 2 animals-14-01800-t002:** Occurrence of clinical adverse effects in the Treatment group (*n* = 17) after the first, second, and third vaccinations. Numbers indicate the serial number of affected mules, or mean ± SD.

Adverse Effects	First Vaccine	Second Vaccine	Third Vaccine
Rectal temperature > 39 °C	6, 10, 18	5, 9, 13, 14, 16, 17	6, 7, 11, 14
	39.5 ± 0.3 °C	39.4 ± 0.4 °C	39.5 ± 0.3 °C
Injection site swelling			
Score 1 (<2 cm)	16	7, 11	12, 15, 18
Score 2 (>2 cm)	-	9, 14, 16	1, 6, 7, 11, 13, 16, 17
Pain response	-	-	-

## Data Availability

The datasets are available from the corresponding author upon reasonable request.

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
