# Peer review of "GnRH Vaccine Could Suppress Serum Testosterone in Stallion Mules"

_animals, 2024, doi:10.3390/ani14121800_

Round 1

Reviewer 1 Report

Comments and Suggestions for Authors

Overview and general recommendation:

The research paper “GnRH vaccine could suppress serum testosterone in stallion mules” reports an alternative to surgical castration in stallion mules using the GnRH vaccine. The vaccine effectiveness has already been tested in horses and donkeys, and the authors report in mules for the first time. The study separated the available animals into groups according to age and if the animal presented cryptorchidism and, besides the number of animals from each group decreased, the differences between the reaction to the vaccine were very interesting. The study also measured the testosterone and the GnRH antibodies, proving the effectiveness of the chosen vaccine dose. Behavior was measured subjectively and only with human handling. This study found a solution for a problem faced by the local military which needs the animals to guarantee the security of the region. They used the animals available and resources to produce a well-designed and well-written paper. The science used to assist current issues and help people is valuable and I congratulate the team of researchers that conducted it.

General concept comments /Major comments:

The dosage used for this experiment was 195 μg per injection per animal. Could you explain the reason you chose this dosage for the mules? Other publications with horses report 400 μg per animal. The authors registered that the application was in the neck area, which is not usual because the equines need to move down their heads during feeding and drinking. The Improvac vaccine generates pain and swollen regions, which could lead to weight loss and not enough drinking because of neck pain. The pectoral muscle is more indicated in this case, avoiding the neck area.

Evaluation of the behavior was only performed with human interaction and not with females in estrous. I understand that the castrations were done to facilitate animal handling. However, a response to a female in heat would be another way to measure the behavior due to vaccination. Were there females available?

Specific comments:

Line 52: Please replace “more or less” with “may”. Suggestion: Besides the stallion mules being infertile, the production of sex hormones is normal, which may influence the stallion's aggressiveness or other unwanted behavior towards humans. 

Line 53: Please replace “be” with “are”

Line 54: For animals the term usually used is “scrotum”, instead of “scrotal sacs”, but this is a suggestion.

Line 55: unsutured

Line 60: Please remove “high risk for”

Line 61: Please replace “any” with “Novel”

Line 106: Regarding the data collection from behavior, the evaluator was the same person during the whole study? Did this person know the animals and their behaviors before starting the study? Please add this information to the methodology. 

Comments on the Quality of English Language

Line 66: sex-hormones. Please fix it throughout the text.

Line 67: “In order to properly manage..” remove “be”

Line 76: has been used..

Line 80: when receiving manipulation from humans

Line 86: Two intact stallions

Line 89: as follows:

Line 96: weeks. When you refer to more than 1 week, it is plural. Please check the whole text.

Line 133: low-concentration

Line 153: due to the limitation of…

Line 188: were not statistically different

Line 188: at a similar age

Line 190: were not statistically different

Line 205: you could replace “manipulation” with “ handling”

Line 225: Clinical Adverse Effects

Line 225: is shown

Line 227: which subsided on the following day.. remove “was”

Line 242: it is only a suggestion: “To the best of my knowledge”

Line 281: remove between, because you already use the word comparing

Line 300: lasted

Line 306: when standing still

Line 318: used in our study

Line 319: compositions

Author Response

Please find the detailed and point-by-point responses to your comments and suggestions in the attached file.

Reviewer 2 Report

Comments and Suggestions for Authors

This is an interesting study however the significance and practicality of reversible immunocastration is lag behind that is at stallion horses. Mules are infertile, not using for breeding if the aggressivity is a problem in working with them the final solution is castration. Anti-GnRH vaccination provides relative short “calm” period and must be repeated several times. But comparing the effect of vaccination on mules and on horses and donkeys is scientifically an interesting research.

 Some details of the study must be clarified:

Line 86-87: Two intact stallion mules were randomly selected for Control-intact. The remaining intact and cryptorchid mules were assigned for Treatment.

- Why did you select only two mules as a control? This is very low individual number of a research group.

Line 95. 195 μg (1.3 mL) of GnRH vaccine (Improvac, Zoetis, 150 μg/mL) intramuscularly at the neck at week 0, 4 and 8.

- Improvac was developed for immunocastration of male pigs. The recommended dose and administration is 400 μg/ pig subcutaneously. Why did the authors choose the dose which they used?

Line 107. Blood samples and the video recordings were collected from each mule every two weeks, starting from two weeks (week -2) before the first vaccination until week 24.

- Serum Testosterone level can show a daily variation. Please provide more details about the time of blood collection (in the morning, in the evening, which time? did collect at the same time of the day?) and duration of video recording. I don’t find any information about the season (date) when the study was performed, I suggest to read „Waddington et al. Testosterone serum profile, semen characteristics and testicular biometry of Mangalarga Marchador stallions in a tropical environment Reprod Domest Anim. 2017 Apr;52(2):335-343. doi: 10.1111/rda.12918.„ Could it be a seasonal changes in tropical conditions in mules?

Line 158. Five mules were indicated as outliers; therefore, statistical analysis was performed based on 20 mules separating into three groups

- What could be the reason for outliers? 5 mules out of 25 are quite many…In which parameters did you find them as outliers? Didn’t they answer for the treatment? Or was the basal Testosterone level much higher?

Line 161. Treatment-cryptorchid-old, n = 1); (b) Control-intact (10-15 years old, n = 2); There are not enough individuals to create a group. At least cryptorchid-old group should be removed from the study as a group and create only one cryptorchid group (n =5).

 Line 158-163: This is connected to M & M part, not to Results.

Fig 3. should be placed between Table 1 and „3.3. Behavior response, changes in total scrotal width and body weight” section for better followability.

Almost all mules had a strong side effect after the vaccine administration. What do you think how can you prevent it? The authors should declare whether recommend this treatment or not for easier handling of stallion mules instead of castration of mules in conclusion part.

Manuscript is recommended for publication after major revision.

Author Response

(The authors gave the same response as above.)

Reviewer 3 Report

Comments and Suggestions for Authors

The manuscript, entitled, “GnRH vaccine could suppress serum testosterone in stallion mules”, describes the results of experimental suppression of testicular function in male mules using GnRH vaccine. The goal of the authors of this work was to perform immuno-castration of the stallion mules resulting in decreased concentration of testosterone leading to easier interactions with humans. While the material presented in this manuscript is interesting and worth presenting to the potential readers, it should not be published in its present form. This paper requires major revisions, including significant improvement in language, clarifying numerous fragments of the paper, redoing statistical analysis, etc.

There are too may language problems to list them all I this review. Therefore, this reviewer suggests that the authors submit this manuscript to a professional editing service before submitting the improved version.

These are selected suggestions and comments:

Summary and abstract

Line 20: “…as a method of immunosuppression…”

Line 21-21: “…testosterone concentration…”

Line 21: “…the effect on the selected behaviors…” or “behavioral characteristics”

Line 21-23: the entire sentence needs to be re-written; it does not read well

Line 36: “…Treatment group…”

Line 38: “…Treatment-young-intact group…”; “…the highest concentration of anti-GnRH antibody…”

Introduction

Line 48-49“…the hybrid between a male donkey (jack) and a female horse (mare)…”; “…used for several types of work…”

Mules do not have military duties but perform specific tasks an army. This needs to be clarified.

Line 49-50: “Mules are intelligent…”, etc. “were” should be replaced by “are”

Line 51: There are reports of fertile mules, which should be mentioned

Line 53-55: “…During routine castrations both testes are surgically removed from the scrotal sacs and the wound is usually left open to drain and heal by the secondary intension…”

Line 56-57: “Retained abdominal testes can produce testosterone in the cryptorchid stallions…”

Line 59-60: What several facilities the authors have in their minds?

Perhaps: “This procedure requires a specialized facility which is usually available in the referral hospitals”

Line 67-72: Not quite clear – needs t be re-written

Line 75: “…There is only one form of the GnRH vaccine that is available in Thailand, a vaccine licensed for male pigs...”

Materials and Methods

Line 85: More information on the cryptorchid mules is needed – all abdominal? Unknown position of the testes? Side of the retained testes?

Line 92: Control-castrated – age?

Line 110-116: More information regarding behavioral observations needed: what is the “restraint stall” – description; what was the experience of all animals used in this study; where the animals led to the “restraint stall”; just before testing/observation, etc. Clear definitions of all terms used for behavioral “grading” needs to be provided. The reader will not know what the term: easy, uneasy, etc., mean.

Line 151-153: Parametric methods of statistical analysis should not be used here due to extremely large variations presented in the results. In some instances, SEM was larger that the mean, and in many instances, SEM was larger that half of the mean. This is not compatible with the parametric analysis, including ANOVA.

Results

General comment: Treatment group or animals, etc., rather than just Treatment, Control-intact, etc. Concentrations should used rather than levels

This chapter is difficult to follow. The authors show different sets of data in various figures, tables and in text. It is my understanding that there were 3 groups, subdivided into subgroups, which gives a total of 6 groups: Control-intact (n=2), Treatment-intact-young (n=4), Treatment-intact-old (n=5), Cryptorchid-young (n=4, Cryptorchid-old (n=1), Control-castrated (n=4). It should be stated when the statistical analysis was done on three groups, with all treatment groups pooled together, and when the analysis was done separately for 6 or 5 groups. Knowing that there is only 1 animal in the Castrated-old group, no statistical analysis could be done with this group as a separate entity. This needs to be clarified. Presenting different sets of data in different figures and the table makes things confusing, especially that similar or the same colors were used for different groups in different figures. Figure 2 should have both control groups as well. Why there is no control castrated group in the table 1? Looks like the Treatment-castrated-old animal was included in the stat analyses, but only if the treatment groups was treated as on group with 14 subjects. This needs to be clarified.

Line 180: Table 1 should be listed as a reference after the first sentence.

Figure 3: the control groups need to be included in this figure

Behavioral observations are difficult to interpret. More details regarding this analysis need to be added – previous experiences, definitions, time of observation, grading (more details than provided so far) in the materials and methods. Also, the progressive conditioning effects might have had confounding effects in all groups. The results do not look convincing enough to state that there was a definite calming effect in the treated animals. This should be discussed later in the discussion section.

Chapter 3.4.

Use of any medications should have been described in the Materials and Methods and further discussed in the results section. How many animals were treated and when? Products, dose, manufacturer, etc. This treatment could have a confounding effects as well.

Discussion – no comments at this point, should be reviewed in details in the revised version.

Comments on the Quality of English Language

There are too may language problems to list them all I this review. Therefore, this reviewer suggests that the authors submit this manuscript to a professional editing service before submitting the improved version.

Author Response

(The authors gave the same response as above.)

Reviewer 4 Report

Comments and Suggestions for Authors

Although the manuscript presents promising results, they are insufficient to consider it for publication. Some issues need to be addressed. On the one hand, the authors have considered 6 animals as outliers. The question is why these animals were discarded. Did they show higher/lower values that made statistics non-significant? Besides, 6 animals out of 25 is a very relevant number. In fact, 24% of the animals have been removed from the statistics. That’s not acceptable. One can think that they have been removed to “arrange” the statistics. On the other hand, group b includes just 2 animals, which is not enough for statistics. Actually, even the 4 animals included in group c are insufficient.

Something else that catches my attention is Figure 1. Despite showing higher values of anti-GnRH antibodies on week 8 concerning week 24, there was no significant difference with control groups. The same event happens on week 16. The SEM is much higher, but still.

Another question that comes to my mind is whether the external testis had been surgically removed in cryptorchid animals. Testosterone values are surprisingly low, which makes me think about that possibility. If the abdominal testis had been removed before the study, these animals shouldn’t be included.

Some small comments follow:

In the chapter on animals, the number of each group must be provided. When the authors talk about the outlier animals, the reader ignores how many animals were previously included in each particular group.  

Since the authors are describing the ELISA protocol, it should be more accurate. The authors comment about the washings, but it’s not specified what washing solution was used: distilled water, PBS, or commercial solution,…

Comments on the Quality of English Language

English must be reviewed. This reviewer has detected some mistakes.

Author Response

(The authors gave the same response as above.)

Round 2

Reviewer 3 Report

Comments and Suggestions for Authors

The manuscript, “GnRH vaccine could suppress serum testosterone in stallion mules”, has been extensively revised and greatly improved. Unfortunately, there are still concerns that need to be addressed before publication of this manuscript. While I am not a native English speaker, I continue to see numerous language errors in the manuscript that need to be corrected. The editor says extensive English editing was performed by MDPI, but I see only a few language corrections in the revised manuscript. Perhaps I did not receive the final version of the manuscript - ? There are still numerous omissions of articles (the, a), not well-constructed sentences, etc.

Another major concern is gathering and analysis of the behavioral data. The authors did not provide the information I was asking for, regarding the previous experiences of the animals of being held in the restraining stocks. Was any of the animals placed in the stocks before this experiment? This is extremely important because all animals should have the same level of prior experience in behavioral research. Also, it is unclear how the behavioral data were analyzed. This is not clarified in the chapter 2.6. Was the analysis (which test?) done within each group separately, only? I am guessing that there was no comparison done between the groups - ? The results of the statistical analysis do not match the impression one has from the data in Table 1. Subjectively, behavioral scores for blood collection decreased in all groups, and in the Treatment group even to a lesser degree than in the other two groups. These changes might have been caused by getting used to the procedure rather than to the vaccine. Perhaps, this problem was caused by large differences in the subjects' numbers between groups. A low number of subjects (mules) in the Control-intact group probably did not allow for statistical differences due to low power statistics. Finally, why only the mean scores were presented in Table 1, while all figures have ±SEM values? Values of SDs, SEMs, or ranges of values (this may be the best choice) should be given. The authors still did not convince me that there was a significant effect of GnRH vaccine on the blood collection scores. I disagree with the statements in lines: 25-27, 44-45, and 500-502.

I am still not clear about the cryptorchids and the castrated mules. The authors should clarify what they know for sure and what they suspect only. Were all mules from the Control-castrated group castrated on both sides? Perhaps, the outlier with a high concentration of testosterone was a cryptorchid? The authors should address this. I am guessing that the exact locations of the retained testes in the unilateral cryptorchids were not determined. This is acceptable, but the authors should make a statement about this: “exact locations of the retrained testes could not be determined.”

Comments on the Quality of English Language

Comments included in the review

Author Response

Please see a point-by-point response to the reviewer’s comments in the attachment.

Reviewer 4 Report

Comments and Suggestions for Authors

The authors have improved the quality of the manuscript. However, some lack remains. The main concern is discussion. The discussion needs to be rewritten. A big part of it consists of repeating the results or adding results not included in the results chapter. For example, from line 395 to line 396. The discussion aims to discuss and compare your results with previous studies and discuss the differences. Similarly, the results need to be developed and justified, not just repeated. As an example: young mule stallions showed higher levels of serum testosterone. How would you explain this result?

On the other hand, some paragraphs are difficult to understand if they are talking about their results or previous studies.

Another example. From lines 400 to 402, the authors state that the differences in serum testosterone may be associated with the temperature and tropical environment. However, in lines 359-360, they state that the effect of the season is probably minimal because Thailand has a similar environment to that of a tropical area. These comments look contradictory.

Last, the discussion must be shortened. Remove the repeated results and focus on discussing the results.

Regarding the conclusions, they must be shortened as well. Once again, the authors are repeating the results, which is not the aim of this chapter.

Some other comments follow:

L59. [EngEdito1]?

L130. [EngEdito2]?

L366-367. Same comment.

L373. Same comment.

L408. Same comment.

Comments on the Quality of English Language

Language has improved

Author Response

(The authors gave the same response as above.)
